# Colorectal Cancer with Peritoneal Metastases: The Impact of the Results of PROPHYLOCHIP, COLOPEC, and PRODIGE 7 Trials on Peritoneal Disease Management

**DOI:** 10.3390/cancers15010165

**Published:** 2022-12-27

**Authors:** Antonio Sommariva, Marco Tonello, Federico Coccolini, Giovanni De Manzoni, Paolo Delrio, Elisa Pizzolato, Roberta Gelmini, Francesco Serra, Erion Rreka, Enrico Maria Pasqual, Luigi Marano, Daniele Biacchi, Fabio Carboni, Shigeki Kusamura, Paolo Sammartino

**Affiliations:** 1Advanced Surgical Oncology Unit, Surgical Oncology of the Esophagus and Digestive Tract, Veneto Institute of Oncology IOV-IRCCS, 35128 Padova, Italy; 2General, Emergency and Trauma Surgery, Pisa University Hospital, 56124 Pisa, Italy; 3Upper GI Surgery Division, University of Verona, 37134 Verona, Italy; 4Colorectal Surgical Oncology, Abdominal Oncology Department, “Fondazione Giovanni Pascale” IRCCS, 80131 Naples, Italy; 5SC Chirurgia Generale d’Urgenza ed Oncologica, AOU Policlinico di Modena, 41124 Modena, Italy; 6General and Peritoneal Surgery, Department of Surgery, Pisa University Hospital, 56124 Pisa, Italy; 7DAME University of Udine-AOUD Center Advanced Surgical Oncology, 33100 Udine, Italy; 8Department of Medicine, Surgery, and Neurosciences, Unit of General Surgery and Surgical Oncology, University of Siena, 53100 Siena, Italy; 9CRS and HIPEC Unit, Pietro Valdoni, Umberto I Policlinico di Roma, 00161 Roma, Italy; 10Peritoneal Tumours Unit, IRCCS Regina Elena National Cancer Institute, 00144 Rome, Italy; 11Peritoneal Surface Malignancies Unit, Fondazione Istituto Nazionale Tumori IRCCS, 20133 Milano, Italy

**Keywords:** peritoneal metastases, colorectal cancer, HIPEC

## Abstract

**Simple Summary:**

Colorectal peritoneal metastases could potentially benefit from locoregional treatments such as cytoreduction combined with hyperthermic intraperitoneal chemotherapy (HIPEC). Three recently published RCTs that have investigated HIPEC in the setting of radical or prophylactic/II^ look surgery have improved the current knowledge of the treatment of CRC PM. The current review summarizes the results of these trials, emphasizing the highlights and criticisms and focusing on the potential impact and future directions in the clinical practice of HIPEC. Oxaliplatin-based HIPEC seems ineffective in improving surgery results in patients with CRC PM. Moreover, the same oxaliplatin-based regimen is ineffective in preventing CRC PM occurrence and should be abandoned. Several ongoing trials are investigating mitomycin-based HIPEC after radical surgery or as a prevention strategy. Meanwhile, HIPEC should still be considered a therapeutic option for selected patients and offered by dedicated, experienced centers and surgical teams.

**Abstract:**

HIPEC is a potentially useful locoregional treatment combined with cytoreduction in patients with peritoneal colorectal metastases. Despite being widely used in several cancer centers around the world, its role had never been investigated before the results of three important RCTs appeared on this topic. The PRODIGE 7 trial clarified the role of oxaliplatin-based HIPEC in patients treated with radical surgery. Conversely, the PROPHYLOCHIP and the COLOPEC were designed to chair the role of HIPEC in patients at high risk of developing peritoneal metastases. Although all three trials demonstrated the relative ineffectiveness of HIPEC for treating or preventing peritoneal metastases, these results are not sufficient to abandon this technique. In addition to some criticisms relating to the design of the trials and their statistical value, the oxaliplatin-based HIPEC was found to be ineffective in preventing or treating peritoneal colorectal metastases, especially in patients already treated with systemic platinum-based chemotherapy. Several studies are ongoing investigating further HIPEC drugs and regimens. The review deeply discussed all the aspects and relapses of this new evidence.

## 1. Introduction

Colorectal cancer (CRC) represents the third most common neoplasm and the third leading cause of death among the population of developed countries [1]. Colorectal peritoneal metastases (PM) are associated with a dismal prognosis compared to patients affected at other metastatic sites [2]. Peritoneal involvement is synchronous with the primary tumor in 5% of newly diagnosed patients and represents 25% of patients with stage IV disease at the onset [3]. After radical surgery, CRC PM are estimated to develop in up to 40% of autoptic studies, even if it became clinically relevant in 15% of cases, suggesting that its real incidence is largely underestimated [4]. Untreated, CRC PM are associated with a survival of less than 9 months, which might be extended to as long as 24 months with systemic chemotherapy [5]. Systemic treatments for CRC PM are proven to be less efficient than those for other metastatic sites (liver, lung, and lymph nodes). The addition of cytoreductive surgery (CRS) combined with hyperthermic intraperitoneal chemotherapy (HIPEC) to standard chemotherapy has been shown to significantly improve the prognosis of patients with CRC PM [6]. In properly selected patients, CRS-HIPEC is associated with a median overall survival of 51 months [7] and can give a potential cure or long-term remission in a significant quote of patients [8]. In a large Dutch population study, a doubling of survival for both PM alone and PM with other involved sites was registered in a period of 20 years (from 6.0 months in 1995–2000 to 12.5 months in 2010–2014, *p* < 0.0001) [9]. The reasons for this improvement are mainly due to more access to systemic treatments and the addition of CRS-HIPEC into the therapeutic pathway of these patients. Even though the indication for CRS-HIPEC is still wrapped in a certain skepticism [10], CRS-HIPEC is included in several oncological guidelines and suggested as professional treatment in selected patients within specialized centers [11]. 

Several questions still remain open, and in the last few years, three important RCTs have been published on the role of HIPEC in the treatment or prevention of CRC PM (Table 1). This narrative review aims to discuss the impact of these studies on CRC PM patients’ management.

## 2. PRODIGE 7 Trial (Is HIPEC Useful in Patients Undergoing Complete Cytoreductive Surgery?)

### 2.1. Trial Design and Results

The UNICANCER PRODIGE 7 trial was designed at 17 centers in France with the main aim of assessing the specific benefit of adding HIPEC to cytoreductive surgery compared with receiving cytoreductive surgery alone [12]. The study randomized 265 patients affected by CRC PM with a peritoneal carcinomatosis index (PCI) < 25 and treated with complete or near complete cytoreduction (<1 mm residual tumor) into two arms (CRS-HIPEC vs. CRS alone) (Figure 1). The HIPEC protocol used in the experimental arm was a bidirectional therapy with intravenous 5-FU and Folinic acid 20 min prior to infusion of intraperitoneal oxaliplatin for 30 min in the perfusion circuit. All patients were included in the study if they completed at least six cycles of systemic chemotherapy before or after surgery. The study’s primary endpoint was overall survival (OS), verifying HIPEC safety. After a median follow-up of 63.8 months, no difference in median OS was detected between the two arms (41.7 months in the cytoreductive surgery plus HIPEC group and 41.2 months in the cytoreductive surgery group, hazard ratio of 1.00, *p* = 0.99). Postoperative mortality and 30-day complication rates were similar between the two arms. However, grade-3 morbidity within 60 days was more frequent in the HIPEC arm (26% vs. 15%, *p* = 0.035). 

This study concluded that no additional overall survival benefit was demonstrated in patients treated with cytoreductive surgery and oxaliplatin-based HIPEC compared with those who received cytoreductive surgery alone to treat peritoneal metastases from colorectal cancer.

### 2.2. Strengths

Even if the PRODIGE 7 trial did not support the use of oxaliplatin-based HIPEC, important points arise from the analysis of the results. Firstly, the trial unequivocally proves the role of cytoreductive surgery in the multimodal treatment of patients with CRC PM. The gain of 42 months in median OS survival is a remarkable result, and a significant quote of patients can be considered cured after surgery (15% at five years). These results are obtained in centers with expertise, following a defined diagnostic and therapeutic pathway and performing surgery according to a standardized surgical plan, including xiphopubic midline laparotomy, complete lysis of adhesion, and full abdominal recesses exploration. After the PRODIGE 7 study, there is a substantial consensus to consider cytoreductive surgery as surgery for CRC liver or lung metastases if adopted in the context of a multimodality approach to systemic chemotherapy after a multidisciplinary discussion of every single case [13,14]. Preoperative staging should be directed toward select patients with limited disease, and the indication for surgery should be balanced with other well-established prognostic factors such as primary tumor presentation (obstruction/perforation), signet ring histology, BRAF and MSI status, and the presence of liver metastases [15,16,17,18,19]. Another important result that should be underlined is the significant effect on the survival of HIPEC in patients with PCI between 11 and 15, although considering the whole population with PCI < 15, this result is not confirmed. 

### 2.3. Limits

Regarding the role of HIPEC in the context of radical cytoreductive surgery, PRODIGE 7 did not support the use of HIPEC after radical surgery. However, many relevant criticisms of this trial emerged. Firstly, the sample size of the trial was targeted at a too optimistic survival benefit (expected gain in survival of 18 months in the HIPEC arm, from 30 to 48 months), with a clear overestimation of this presumed benefit. This choice might be partly influenced by the need to recruit a sufficient number of patients for randomization. The role of HIPEC itself in treating optimally resected colorectal PM has never been investigated before the PRODIGE 7 study. A prospective randomized trial comparing early postoperative intraperitoneal chemotherapy (EPIC) plus systemic chemotherapy with systemic chemotherapy alone, both after complete cytoreductive surgery of colorectal PM, was stopped prematurely [20]. This latter study confirmed the difficulties in conducting randomized trials in this subset of patients (isolated PM with PCI less than 20). Moreover, the choice of overall survival as the primary endpoint does not adequately emphasize the role of CRS-HIPEC, which is primarily a locoregional treatment directed to control metastatic disease inside the peritoneal cavity. Only a minority of patients will be cured after CRS-HIPEC, and survival results are mainly related to response to systemic chemotherapy. For this reason, peritoneal recurrence/progression-free survival would have been more appropriate as the main aim of the study.

Another relevant point is the option for the patients recruited in the CRS-only arm to be treated with CRS-HIPEC in case of peritoneal recurrence. This cross-over option was chosen by 12% of patients (16 patients), and it is complicated to address its contribution in the final result of the trial, even if the survival results are the same considering per protocol populations (which exclude cross-over patients) and intention to treat populations analysis. Again, recruitment concerns have prompted the investigators to offer this option to the patients. 

The extent of PM is a well-known factor for selecting patients with a potential benefit from CRS-HIPEC and is generally expressed by the peritoneal cancer index (PCI). A PCI of less than 25 was one of the inclusion criteria for the trial. However, a high proportion of patients (84%) underwent preoperative systemic chemotherapy, and no data are available on PCI before starting neoadjuvant treatments. Presumably, the chemotherapy has favored, at the time of randomization (i.e., at the time of CRS-HIPEC), the inclusion of a significant number of patients with a PCI higher than 20, accordingly with no benefit from CRS-HIPEC. Considered that PCI is the main prognostic factor for patients with CRC PM selected for surgery, PCI assessment before neoadjuvant chemotherapy and stratification based on PCI would have possibly influenced the results of PRODIGE 7.

Other critical points of the PRODIGE 7 are the potential effect of perioperative chemotherapy (both groups have received a median of six cycles of chemotherapy) and the efficacy of 30 min of oxaliplatin HIPEC. More than three-quarters of the patients (85%) have received oxaliplatin-based neoadjuvant chemotherapy (FOLFOX). In a recent study, oxaliplatin chemoresistance was demonstrated in the preoperative setting, potentially making the HIPEC regimen ineffective [21]. Patients who received oxaliplatin before CRS-HIPEC had significantly altered chemosensitivity to this drug. In an experimental model, peritoneal nodules obtained during surgery and tested for in vitro growth inhibition with oxaliplatin, mitomycin C, 5-FU, and irinotecan (0.03 μg/mL) showed different responses; only 20 of 51 patients who received oxaliplatin-based NACT were sensitive to oxaliplatin, compared with 16 of 24 who did not have oxaliplatin-based NACT (*p* = 0.046). In contrast, patients treated with or without oxaliplatin-based NACT had similar results in terms of chemosensitivity to MMC (20 of 40 versus 11 of 18; *p* = 0.571). Oxaliplatin chemoresistance was confirmed by an ex-vivo analysis of programmed cell death (EVA/PCD) in colorectal cancer cells obtained from patients evaluated for chemotherapy sensitivity [22]. CRC cells from previously untreated patients had a statistically significant lower lethal concentration of 50% (LC50) for oxaliplatin (*p* = 0.002) compared to the LC50 for previously treated cancer cells. There was a borderline statistical difference in LC50 between treated and untreated cancer cells (*p* = 0.066) when they were exposed in vivo to 5-fluorouracil. In testing irinotecan or mitomycin C, no significant difference in the LC50 value was detected between treated and untreated colon cancer cells [23]. These studies raise the question of whether systemic oxaliplatin might induce chemoresistance and influence the efficacy of oxaliplatin-based HIPEC in the PRODIGE 7 trial.

Regarding the efficacy of an oxaliplatin-based protocol, data on the best HIPEC protocol for CRC PM are controversial, and a meaningful comparison between the two most used HIPEC regimens is difficult [24]. Experimental data obtained from organoids, either from appendiceal or colon organoids, treated in parallel at 37 and 42 °C with MMC (40 mg/3 L over 2 h), oxaliplatin (200 mg/m^2^ over 2 h), or oxaliplatin (460 mg/m^2^ over 30 min), showed similar viability between MMC and 200 mg/m^2^ oxaliplatin. In contrast, MMC was more effective in comparison to a 460 mg/m^2^ oxaliplatin treatment (27% vs. 53%, *p* = 0.002) [25].

Besides its potential inefficacy, the trial also showed that oxaliplatin-based HIPEC is related to a higher rate of postoperative complications (grade-3 morbidity in the HIPEC arm was 26% versus 15%, *p* = 0.035) and, in general, to a longer postoperative course (18 days versus 13, *p* = 0.0001) and time to resumption of postoperative systemic chemotherapy. This morbidity profile of the trial tended to be more medical than surgical (intra-abdominal 6% versus 3% and extra-abdominal 21% versus 14% in the HIPEC versus no-HIPEC arms, respectively). Two recently published meta-analyses confirm a significantly higher rate of major complications after oxaliplatin-based HIPEC compared to MMC-based HIPEC, even if its role as a contributory factor remains controversial and difficult to estimate due to the heterogeneity of the studies included in the analysis [24,26]. 

## 3. PROPHYLOCHIP-PRODIGE 15 (Is Second-Look Surgery and HIPEC Useful in Patients at High Risk of Peritoneal Recurrence?)

### 3.1. Trial Design and Results

The PROPHYLOCHIP trial is an open-label, randomized, phase 3 study performed in 23 centers in France that has investigated the potential role of second-look surgery associated with HIPEC in patients with colorectal cancer considered at high risk of peritoneal recurrence (Figure 2) [27]. Patients considered at risk for peritoneal relapse and included in the trial were those with synchronous PM or ovarian metastases radically resected at the time of treatment of the primary or perforated tumors. All the patients (*N* = 150) received six months of systemic chemotherapy, and if without evidence of disease recurrence on CT scan, they were randomized for second-look surgery or surveillance. Second-look surgery consisted of an exploratory laparotomy, CRS if PM was present, followed by HIPEC in all the patients according to different schedules (oxaliplatin 460 mg/m², or oxaliplatin 300 mg/m² plus irinotecan 200 mg/m², plus intravenous fluorouracil 400 mg/m²), or mitomycin-HIPEC (mitomycin 35 mg/m²) alone in case of neuropathy intraperitoneal oxaliplatin for 30 min and intravenous 5-FU immediately before starting the HIPEC. The primary outcome was 3-year DFS (disease-free survival), defined as the time from randomization to peritoneal or distant disease recurrence or death from any cause. The result of the trial was negative, as no difference in 3-year DFS between the experimental (II^ look + CRS-HIPEC) and surveillance arms (44% vs. 53%, respectively; hazard ratio 0.97, 95% CI: 0.61–1.56, *p* = 0.82) was detected. There was also no difference in terms of 5-year overall survival between the two arms of the study (second-look surgery versus surveillance, 68% and 72%, respectively). Besides the negative result and the absence of postoperative mortality, the morbidity rate in the experimental arm was quite significant, with a major postoperative complication rate (grades 3–4) of 41% (29 over the 71 patients), especially in the abdomen (12%). In eight patients (11%), treatment of intra-abdominal complications required a reoperation.

The conclusion of the trial is that a systematic second-look surgery plus oxaliplatin-based HIPEC did not improve disease-free survival compared with standard surveillance and was burdened by a significant grade of serious complications.

### 3.2. Strengths

Although the second-look strategy tested in the PROPHYLOCHIP trial did not improve disease-free survival, this study highlighted the value of defined risk factors for peritoneal recurrence, particularly peritoneal metastases resected at the time of treatment of the primary. The occurrence of peritoneal metastases during II^ look exploration is as high as 50%, with a median PCI less than 14 in around three-fourths of the patients explored, confirming the low sensitivity of standard imaging in detecting peritoneal recurrence, even in high-risk patients.

### 3.3. Limits

The study suffers from the same limitations and doubts highlighted above for the PRODIGE 7 study. The oxaliplatin HIPEC regimen is probably not the most effective in treating peritoneal implants. Moreover, almost 90% of the patients had been treated before the operation with an oxaliplatin-based regimen of systemic chemotherapy, which could have induced a sort of chemoresistance to HIPEC. Another limitation is that the primary outcome was analyzed by intention to treat. All the patients were managed in centers with experience in the diagnosis and treatment of peritoneal disease, and a large proportion of patients undergoing surveillance may have been treated at an early stage, minimizing the benefit of the II^ look strategy. Finally, synchronous PM removed at the time of the primary tumor treatment is a risk factor with a completely different relevance on prognosis compared to the T4 tumor. It is challenging to consider second-look surgery and HIPEC as a prophylactic strategy in patients with already PM at presentation.

## 4. COLOPEC (Is Prophylactic HIPEC Useful in Patients at High Risk of Peritoneal Metastases?)

### 4.1. Trial Design and Results 

COLOPEC was a multicenter, open-label trial phase III RCT done in specialized centers for HIPEC in the Netherlands [28]. The trial evaluated adjuvant and prophylactic HIPEC in patients with advanced colon cancer (T4NxM0) or perforated disease without PM (Figure 3). Patients were randomized before resection of the primary tumor to adjuvant HIPEC followed by adjuvant systemic chemotherapy (experimental group) or adjuvant systemic chemotherapy alone (control group). Adjuvant HIPEC (oxaliplatin 460 mg/m^2^ for 30 min at 42 °C and 5FU, 400 mg/m^2^, and leucovorin, 20 mg/m^2^ intravenously) has been delivered at the time of primary tumor resection or within 5–8 weeks after. The primary endpoint was 18-month peritoneal metastases-free survival, evaluated by a diagnostic laparoscopy performed on patients who showed no signs of peritoneal recurrence at 18 months. During the study, 19 patients (19%) developed PM: eight cases during follow-up, nine during surgical exploration, and two at the 18-month laparoscopy. In the intention-to-treat analysis, no difference in 18-month peritoneal-free survival was detected between groups 80.9% (95% confidence interval [CI] 73.3–88.5 for the experimental arm versus 76.2% (95% CI 68–84.4) for the control arms, log-rank two-sided *p* = 0.28). Peritoneal metastases were treated with cytoreductive surgery plus HIPEC in 13 (68%) out of 19 patients in the experimental group and 15 (65%) out of 23 patients in the control group.

### 4.2. Strengths

The COLOPEC study confirms that in patients with locally advanced CRC (T4 or perforated), the risk of peritoneal relapse over the 3 years of follow-up in the overall study population is 21%. This data confirms previous reports and clearly indicates the relevance of the clinical problem in this setting of patients.

### 4.3. Limits

Besides the already discussed doubts on the efficacy of oxaliplatin-based HIPEC, a significant rate of patients (around 10%) were found to have peritoneal disease at early exploration that was not identified at the time of the primary treatment or might represent an early progression secondary to the primary cancer manipulation. The significant rate of undetected PM might have altered the trial’s design and obscured the potential effect of adjuvant HIPEC. Another limitation is that in the experimental arm (prophylactic HIPEC), adjuvant chemotherapy was administered later in the experimental group than in the control group. Although the rate of patients receiving adjuvant treatment did not differ significantly between groups, the time for starting chemotherapy was longer in the experimental group than in the control group (6 weeks [IQR 5–7] versus 10 weeks [9,10,11,12]; *p* < 0.0001). This delay might be partly caused by the higher morbidity rate observed in the experimental arm. In patients treated with simultaneous HIPEC, major complications (grades 3–4) occurred in 37.5% of cases (3 out of 8), with an overall complication rate of 88%. In the staged HIPEC group, the overall morbidity was lower (6%), with a reintervention rate of 3%. This might have altered the natural course of disease in patients treated with prophylactic HIPEC. 

Moreover, diagnostic laparoscopy was revealed to be a too invasive method to determine the primary endpoint of the study, reflected by the fact that only 63% of patients underwent an 18-month diagnostic laparoscopy. Although in both groups, the proportion of patients undergoing 18-month diagnostic laparoscopy was similar and likely equally distributed between the groups, more than one-fourth of the patients might have missed PM at the time of laparoscopic evaluation. Furthermore, it is well known that laparoscopy is less sensitive than laparotomy for diagnostic and staging purposes, another reason for potentially affecting the primary endpoint of the study.

## 5. The Impact of Trials and Future Directions

The results of PROPHYLOCHIP, COLOPEC, and mostly of the PRODIGE 7 trials have been considered as the end of HIPEC and its rationale in CRC PM management [29]. However, all the trials tested a single drug regimen (oxaliplatin) and HIPEC duration (30 min) in a selected group of patients. The debate on the role of HIPEC in CRC PM treatment is constantly in evolution [30]. In a survey of expert surgeons of the PSOGI investigating the impact of the results of PRODIGE 7 on the clinical management of CRC PM, the trials seem to have had a major impact on the practice of CRS-HIPEC for CRC PM [31,32]. The panelist considers CRS-HIPEC still an indication for CRC PM. However, the national consensus for CRS-HIPEC has decreased in most countries (10 over 19), and in two countries, CRS-HIPEC was removed from the national guidelines. The most interesting result of the survey was the change of HIPEC regimen in most of the panelists, with a shift toward MMC-based HIPEC and a partial discontinuation of oxaliplatin HIPEC regimens or its stop as monotherapy.

As reported above, one of the major issues requiring further investigation after the PRODIGE 7 trial publication (and COLOPEC and PROPHYLOCHIP as well) is the choice of intra-peritoneal or combination of drugs for HIPEC in CRC PM. The critical point in discussing HIPEC efficacy is the oxaliplatin-based protocol, which is the same one used in the three trials. The results of these trials raised the question of whether intraperitoneal oxaliplatin under hyperthermia (460 mg/m^2^ for 30 min) should not be further used, as it probably does not represent the best choice in the treatment of microscopic disease either after cytoreductive surgery and with prophylactic intent. 

Besides the potential impact of intraperitoneal oxaliplatin chemoresistance induced by the neoadjuvant oxaliplatin systemic chemotherapy adopted in the PRODIGE 7 protocol, it remains unclear whether a different HIPEC protocol, different drugs or drug combinations might potentially change the obtained survival results [33,34,35,36,37]. HIPEC should not be considered ineffective until new protocols, new drugs, or drug combinations have been evaluated in well-designed comparative studies. In this perspective, several strategies have been proposed as a rationale for future trials. Considering that up to one-half of oxaliplatin is systemically absorbed over 30 min of HIPEC, thus limiting its exposure to the peritoneal layers, extending the HIPEC time up to 120 min with a lower drug dose could improve the oxaliplatin-based HIPEC efficacy. In future trials, the addition of a second drug to oxaliplatin (i.e., irinotecan) or shifting to a mitomycin C-based HIPEC or mitomycin C combined with cisplatin could also potentially overcome the clinical (and experimentally proven) induced chemoresistance of neoadjuvant systemic oxaliplatin [38,39]. A trial is planned within the French Institution with a similar design of the PRODIGE 7 with a mitomycin C-based HIPEC. A list of ongoing trials investigating prophylactic HIPEC/II^ Look protocols in CRC PM is reported in Table 2.

The results of the Spanish trial HIPECT4 (ClinicalTrials.gov NCT02614534) have been recently presented at ESMO 2022 (abstract n 314O) [40]. Proactive cytoreductive surgery associated with mitomycin C-based HIPEC at the time of primary cT4 colorectal cancer curative resection leads to a significant improvement in the 3-year locoregional control rate (97% in the HIPEC group vs. 87%). However, the DFS was not statistically different between groups.

## 6. Conclusions

The results of recently published oxaliplatin HIPEC regimens have created a certain skepticism in the cancer community regarding the therapeutic potential of HIPEC in CRC PM treatment. Further high-quality studies are needed for testing new HIPEC regimes. Additionally, the II^ look/prophylactic strategies, besides the HIPEC regime adopted, do not seem to add any benefit to CRC patients at risk of PM, and more effective methods (radiological, biomarkers) for PM early detection in surgically treated patients are warranted.

## Figures and Tables

**Figure 1 cancers-15-00165-f001:**
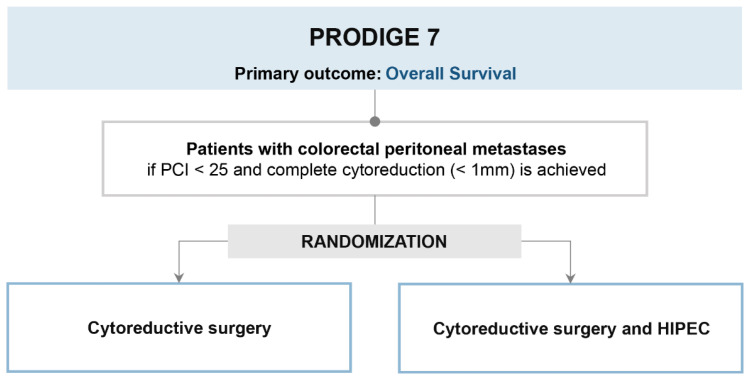
PRODIGE 7 trial design.

**Figure 2 cancers-15-00165-f002:**
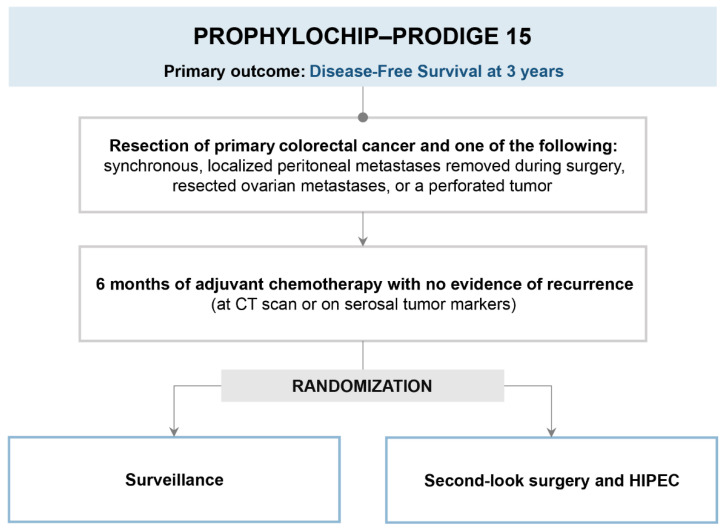
PROPHYLOCHIP-PRODIGE 15 trial design.

**Figure 3 cancers-15-00165-f003:**
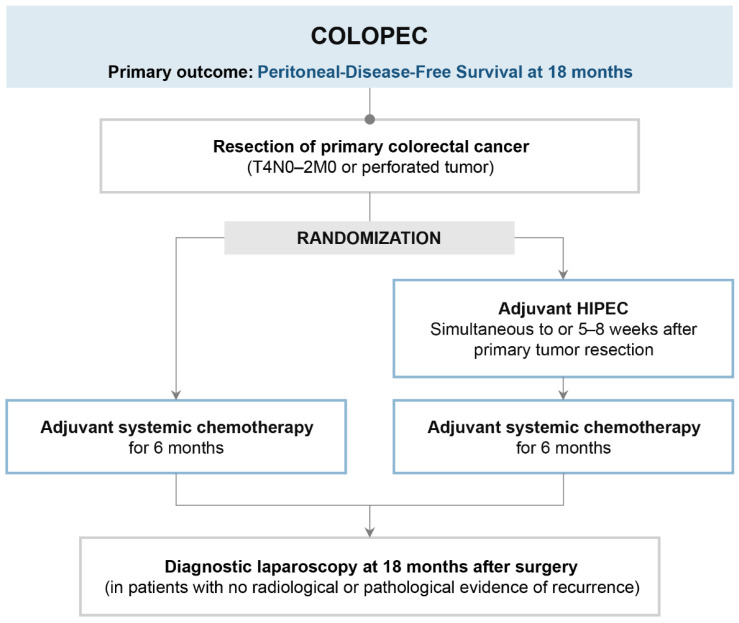
COLOPEC trial design.

**Table 1 cancers-15-00165-t001:** Key points of PRODIGE 7, PROPHYLOCHIP-PRODIGE 15, and COLOPEC trials.

Trials	Type	Setting	Patients	Arms	Primary End Point	Results
PRODIGE7 (France)	Multicenter	Curative	PM with PCI < 20	CRS alone versus CRS-HIPEC	OS (months)	41.2 vs. 41.7 *p* = NS
PROPHYLOCHIP-PRODIGE 15 (France)	Multicenter	Prophylactic	High risk (peritoneal/ ovarian metastases radically resected/perforated tumor) after systemic CHT	Follow-up versus CRS-HIPEC	3-years DFS (Peritoneal or not)	53% vs. 44% *p* = 0.82
COLOPEC (The Netherlands)	Multicenter	Adjuvant	High risk (T4/perforated tumor)	Systemic CHT alone versus Systemic CHT + HIPEC	Peritoneal-Disease -Free Survival (at 18 months)	76.2% vs. 80.9% *p* = 0.28

**Table 2 cancers-15-00165-t002:** Randomized controlled trials investigating prophylactic HIPEC/II^Look in CRC PM.

Sponsor	Type	Setting	ClinicalTrials.gov Identifier:	Population	Experimental Arm	Primary Outcome
Cordoba (Spain)	Multicenter	Prophylactic	NCT02614534	Primary cT4 colorectal cancer undergoing curative resection	Proactive cytoreductive surgery + HIPEC (mitomycin C 30 mg/m^2^)	DFS
Zhejiang University (China)	Multicenter	II^look	NCT02179489	Resected Minimal Synchronous PC or Ovarian Metastases, tumor rupture or identified cT4 after 6 months of systemic chemotherapy	II^ look surgery and HIPEC with MMC (mitomycin C 30 mg/m^2^)	DFS
Policlinico Umberto I, Sapienza University, Rome (Italy)	Multicenter	Prophylactic	NCT02974556	Primary cT3/4 colorectal cancer undergoing curative resection	Proactive cytoreductive surgery + HIPEC (oxaliplatin 260 mg/m^2^ + 5-FU 400 mg/m^2^ and leucovorin 20 mg/m^2^ i.v.)	DFS
Guangzhou Medical University, Beijing (China)	Multicenter	Prophylactic	NCT04370925	Primary cT4 colorectal cancer	Proactive mitomycin c (30 mg/m^2^)	DFS
Policlinico Universitario Gemelli, Rome (Italy)	Multicenter	Prophylactic	NCT03914820	Primary cT4 colorectal cancer with or without perforation, minimal and limited peritoneal metastasis in close proximity to the primary tumor, ovarian metastases undergoing radical resection	HIPEC (mitomycin C 35 mg/m^2^)	LDFS
Hospital Universitario de Fuenlabrada (Madrid, Spain)	Multicenter	Curative	NCT05250648	PM with PCI less than 20 completely resected (no macroscopic residual disease)	cytoreductive surgery + HIPEC (mitomycin C 35 mg/m^2^)	PRFS

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
