# Peer review of "Colorectal Cancer with Peritoneal Metastases: The Impact of the Results of PROPHYLOCHIP, COLOPEC, and PRODIGE 7 Trials on Peritoneal Disease Management"

_cancers, 2022, doi:10.3390/cancers15010165_

Round 1

Reviewer 1 Report

The authors have written a manuscript reflecting on the latest three HIPEC trials. I congratulate them with the relevant remarks about strengths and remarks of the three studies. The trials being the PRODIGE 7,  PROPHYLOCHIP and  COLOPEC, have not shown attributive effect of hipec and adjuvant hipec with oxaliplatin. The strength of this review lies in the emphasis of the shortcoming of the early trials and highlighting the potential impact of intraperitoneal oxaliplatin chemo resistance induced by the neoadjuvant oxaliplatin systemic chemotherapy

I do agree with the authors that these trials are not sufficient conclude that HIPEC is not effective ; HIPEC with mitomycin with out prior systemic chemotherapy is still considered a potential curative treatment. In addition precise cytoreduction can only be offered by dedicated experienced centers/surgical teams and is a substantial part of the treatment. The danger is that   oncologists and surgeons may refrain from offering cytoreductive surgery / hipec because of these recent discussions

Nevertheless; scientifically it is neither a systematic review nor a future outlook nor opinion text . IN addition it lacks robust discussion of the future trials. The conclusion could put more emphasis, in addition to the discussion, on the relatively singular type of HIPEC protocols and beforementioned potential therapy resistance in the three reviewed trials and the possibly preliminary elimination of all CRS-HIPEC in some national guidelines.

A table giving a short summary of the included trials (PROPHYLOCHIP, COLOPEC and PRODIGE 7) with their respective key features and shortcomings would be an improvement of the readability of the review. The addition of second look trials convolutes the review by mixing treatment and diagnostic oriented trials and could be considered for a separate review.

Author Response

The authors have written a manuscript reflecting on the latest three HIPEC trials. I congratulate them with the relevant remarks about strengths and remarks of the three studies. The trials being the PRODIGE 7,  PROPHYLOCHIP and  COLOPEC, have not shown attributive effect of hipec and adjuvant hipec with oxaliplatin. The strength of this review lies in the emphasis of the shortcoming of the early trials and highlighting the potential impact of intraperitoneal oxaliplatin chemo resistance induced by the neoadjuvant oxaliplatin systemic chemotherapy

Thank you for the comment

I do agree with the authors that these trials are not sufficient conclude that HIPEC is not effective ; HIPEC with mitomycin with out prior systemic chemotherapy is still considered a potential curative treatment. In addition precise cytoreduction can only be offered by dedicated experienced centers/surgical teams and is a substantial part of the treatment. The danger is that   oncologists and surgeons may refrain from offering cytoreductive surgery / hipec because of these recent discussions

We agree. For these reasons, we have emphasized that the role of HIPEC is not at the end and further evidence is needed. 

Nevertheless; scientifically it is neither a systematic review nor a future outlook nor opinion text . IN addition it lacks robust discussion of the future trials. The conclusion could put more emphasis, in addition to the discussion, on the relatively singular type of HIPEC protocols and beforementioned potential therapy resistance in the three reviewed trials and the possibly preliminary elimination of all CRS-HIPEC in some national guidelines.

 Thank you for your comments. We agree that this review represents something unconventional. However, for surgeons and oncologists dealing with peritoneal metastases, our text offers a short and complete summary of the strengths and remarks of these important trials. We have arranged the text considering your comments, putting more emphasis in the discussion section of the drawback of oxaliplatin chemoresistance and the potential ways to overcomes this problem. Several studies are still ongoing e no further conclusion can be made at this moment. As reported in the conclusion, HIPEC should not be abandoned and validation of new drugs protocols are needed.  We have also included a table which summarize the key points of the three trials.

A table giving a short summary of the included trials (PROPHYLOCHIP, COLOPEC and PRODIGE 7) with their respective key features and shortcomings would be an improvement of the readability of the review.

We have included a table with the key points and results of the three trials

The addition of second look trials convolutes the review by mixing treatment and diagnostic oriented trials and could be considered for a separate review.

Our review is focused on the RCTs (published and ongoing) investigating HIPEC in the treatment of CRC PM either after curative cytoreduction surgery either in the setting of a prophylactic/II look approach

Reviewer 2 Report

Meanwhile the p.o. morbidity and mortality is referred for Prodige 7 trial, no data  of these complications is reported in the paper  for the others trials in which only one  arm of the study involves employment of HIPEC.  These data can help in the critical analysis of the trials.

Author Response

Meanwhile the p.o. morbidity and mortality is referred for Prodige 7 trial, no data  of these complications is reported in the paper  for the others trials in which only one  arm of the study involves employment of HIPEC.  These data can help in the critical analysis of the trials.

Thank you for your suggestion. We have included these data in the appropriate section.